# Polyamines in Edible and Medicinal Fungi from Serbia: A Novel Perspective on Neuroprotective Properties

**DOI:** 10.3390/jof10010021

**Published:** 2023-12-28

**Authors:** Milena Rašeta, Marko Kebert, Jovana Mišković, Milana Rakić, Saša Kostić, Eleonora Čapelja, Maja Karaman

**Affiliations:** 1Department of Chemistry, Biochemistry and Environmental Protection, Faculty of Sciences, University of Novi Sad, Trg Dositeja Obradovića 3, 21000 Novi Sad, Serbia; 2Institute of Lowland Forestry and Environment, University of Novi Sad, Antona Čehova 13d, 21000 Novi Sad, Serbiasasa.kostic@uns.ac.rs (S.K.); 3ProFungi Laboratory, Department of Biology and Ecology, Faculty of Sciences, University of Novi Sad, Trg Dositeja Obradovića 2, 21000 Novi Sad, Serbiamilana.novakovic@dbe.uns.ac.rs (M.R.); eleonora.capelja@dbe.uns.ac.rs (E.Č.); maja.karaman@dbe.uns.ac.rs (M.K.)

**Keywords:** polyamines, *G. applanatum*, *L. nuda*, antioxidant, phenolics, synergism, anti-acetylcholinesterase

## Abstract

The therapeutic effectiveness of current neurodegenerative disease treatments is still under debate because of problems with bioavailability and a range of side effects. Fungi, which are increasingly recognized as sources of natural antioxidants and acetylcholinesterase (AChE) enzyme inhibitors, may thus serve as potent neuroprotective agents. Previous studies have associated the anti-AChE and antioxidant activities of fungi mostly with polysaccharides and phenolic compounds, while other secondary metabolites such as polyamines (PAs) have been neglected. This study aimed to investigate eight edible and medicinal fungi from Serbia, marking the initial investigation into the neuroprotective capabilities of *Postia caesia*, *Clitocybe odora*, *Clitopilus prunulus*, and *Morchella elata*. Neuroprotective activity was examined using the Ellman assay, while the antioxidant capacity was tested by conducting DPPH, NO, ABTS, and FRAP tests. PA levels were determined by high-performance liquid chromatography (HPLC) coupled with fluorescent detection. *Ganoderma applanatum* and *Lepista nuda* exhibited the most robust anti-AChE (98.05 ± 0.83% and 99.94 ± 3.10%, respectively) and antioxidant activities, attributed to the synergistic effects of the total protein, total phenolic, and PA levels. Furthermore, *P. caesia* displayed significant AChE inhibition (88.21 ± 4.76%), primarily linked to the elevated spermidine (SPD) (62.98 ± 3.19 mg/kg d.w.) and putrescine (PUT) levels (55.87 ± 3.16 mg/kg d.w.). Our results highlight the need for thorough research to comprehend the intricate relationships between distinct fungus species and AChE inhibition. However, it is important to recognize that more research is required to identify the precise substances causing the reported inhibitory effects.

## 1. Introduction

Neurodegenerative diseases (NDDs) represent a diverse group of pathological conditions characterized by the progressive and irreversible degeneration of nervous tissue, primarily affecting the elderly. Despite the clinical heterogeneity of neurological diseases, such as Alzheimer’s (AD) and Parkison’s, elevated levels of oxidative stress have been consistently identified as a prominent hallmark in their pathogenesis [1,2,3]. A demographic projection that the global population aged over 60 will surpass 2 billion within the next 30 years underscores the challenge posed by the absence of effective treatments for NDDs, as the increasing elderly population intensifies the socioeconomic impact of these conditions [2]. When it comes to AD, an association of the degeneration of cholinergic and dopaminergic neurons with low levels of acetylcholine led to the creation of the “cholinergic hypothesis”, where inhibitors of the acetylcholinesterase (AChE) enzyme have been used to raise the level of acetylcholine [4]. However, despite animal models consistently showing increased brain acetylcholine and improved cognitive deficits in AD, the clinical effectiveness of AChE inhibitors, including natural alkaloids huperzine A and galantamine, remains a subject of debate due to challenges related to bioavailability and various side effects [3,4,5].

As a result, fungi have gained recognition as sources of non-alkaloid agents that inhibit the AChE enzyme, and they are also acknowledged for being organisms that naturally contain antioxidants [3]. Nevertheless, edible mushrooms are considered nutritionally valuable since they are low in calories (with low fat content) and rich in minerals (especially potassium), essential amino acids, vitamins (including provitamin D_2_ and vitamin B_12_), and fiber [6]. Additionally, mushrooms contain various natural compounds with diverse positive health effects and medicinal benefits, including positive impacts on brain health and cognition, weight management, oral health, constipation, and diabetes [7]. Many of the immunomodulatory effects associated with mushrooms are linked to polysaccharides, specifically β-glucans or polysaccharide–protein complexes [3], while a portion of their therapeutic properties is related to their ability to inhibit oxidative stress [7] which is attributed to the presence of phenolics [3,8,9,10] and bioactive amines [6,11,12].

Polyamines (PAs) represent a ubiquitous category of small, positively charged polycations within the broader spectrum of biogenic amines [13,14]. They play a crucial role in various physiological processes, including but not limited to cell growth, gene regulation, autophagy, differentiation, proliferation, the modulation of ion channels, the regulation of inflammation, oxidative stress management, and immune responses [1,13,15]. Triamine spermidine (SPD), tetraamine spermine (SPM), and diamine precursor putrescine (PUT) are the most common natural amines found extensively in all cells, but variations in PA concentrations have been observed in fruits, vegetables, and animal-derived food products [2]. The levels of individual PA levels depend on the coordination of biosynthesis, degradation, uptake, and excretion reactions, since they are highly interconvertible, and deviations from the narrow concentration range can lead to significant physiological and metabolic disturbances [14]. A substantial decrease in PA levels can impede cell proliferation and migration, while conversely, an excess of polyamines leads to apoptosis and cell transformation [13]. PA degradation is a prooxidative process, since it leads to the generation of reactive aldehydes and reactive oxygen species (ROS) which contributes to increased oxidative stress, partly attributed to the reduction in PA levels, while, on the other hand, PAs are potent antioxidants and ROS scavengers [14].

Nevertheless, the decrease in PA levels with age has been a long-recognized phenomenon, which leads to the age-related impairment of cognitive and other behavioral reactions [1]. This observation prompts the hypothesis that preserving SPD levels during the aging process may contribute to an extension of longevity and that PA levels can be increased through supplementation in food or water [16,17].

Examples of food and beverages rich in PAs include rice bran, green pepper, broccoli, soybeans, mushrooms, oranges, and green tea, which opens avenues for exploring the potential correlation between PA intake and the aging process, providing a practical approach to studying their effects on longevity and antioxidant activity as well [1,15,16,17]. Studies have demonstrated that SPM, SPD, and PUT effectively act as free radical scavengers, bind to toxic aldehydes, and showcase chelating abilities toward metal ions, while SPM and SPD exhibit the capacity to modify the activity of antioxidant enzymes through their chaperone-like activity and through the alternation of gene expression, contributing to the modulation of oxidative stress-triggering mechanisms [1,11,18,19]. Given these critical roles, there is currently a special emphasis on antioxidant PAs and phenolic compounds as well as polyamines conjugated with phenolics such as phenolamides in nutrition science and, preferably, those secondary metabolites that can prevent harmful environmental and dietary effects, while amidst the myriad sources of these intriguing molecules, edible and medicinal fungi emerge as particularly promising reservoirs.

The advantageous effects of fungal polyphenols are mainly connected to their antioxidant attributes, whereas carbohydrates obtained from fungi exhibit immunomodulatory activity, along with various other biological activities. Nevertheless, emerging studies indicate that these compounds might also contribute to promoting neuroprotection [8,20,21,22]. Generally, previous findings have emphasized the potential of phenolic acids from fungi—particularly caffeic, cinnamic, and *p*-hydroxybenzoic acids—as effective antioxidants and inhibitors of AChE [3,8,22,23], while the effects of other secondary metabolites (e.g., polyamines, terpenes, etc.) on the mentioned activities are largely neglected.

There are limited data on the PA contents in edible mushrooms, with an emphasis on cultivated mushrooms such as *Agaricus bisporus*, *Flammulina velutipes*, *Lentinula edodes*, *Grifola frondosa*, *Hericium erinaceus*, *Pleurotus ostreatus*, *Volvariella volvacea*, etc. [6,24]. The predominant PAs found in fungi are PUT, SPD, and SPM; however, SPM is not universally present in all fungi [14].

Since the prevalence of neurodegenerative disorders continues to rise globally, the identification and characterization of novel neuroprotective agents from natural sources holds significant promise for the development of innovative therapeutic strategies.

This study is focused on exploring the occurrence and quantification of main polyamines and its correlation to the antioxidant and neuroprotective activities of eight edible and medicinal fungi from Serbia. Hence, this pioneering report is aimed at unravelling the complex interaction between PAs, phenolics, neuroprotection, and antioxidant activity in autochthonous wild fungal species (*Lepista nuda*, *Clitocybe odora*, *Clitopilus prunulus*, *Cyclocybe aegerita*, *Lepista nuda*, *Morchella elata*, *Ganoderma applanatum,* and *Ganoderma resinaceum*), shedding light on their potential role in mitigating neurodegenerative processes.

## 2. Materials and Methods

### 2.1. Fungal Material

Fruiting bodies of eight selected fungal species were collected at full maturity from different regions of Serbia and were identified based on their microscopic and macroscopic characteristics by Dr. Eleonora Čapelja. They were stored and deposited at the ProFungi Laboratory, Department of Biology and Ecology, University of Novi Sad (Serbia), under the numbers presented in Table 1.

The fruiting bodies of the selected fungal species were sliced into small fragments and subsequently subjected to hot-air drying in an oven (Universal oven UF55—Memmert GmbH + Co. KG, Schwabach, Germany) at 45 °C for 24 h prior to additional analysis. All samples were kept in a light-protected storage within paper bags.

### 2.2. Polyamine Analysis

Fungal tissues, approximately 30 mg of lyophilized material (dry weight—d.w.), underwent extraction using a solution of 4% perchloric acid (*v*/*v*) in a volume ten times that of the sample. The resultant homogenate was cooled on ice for 1 h and then centrifuged at 15,000× *g* for 30 min. Following this, supernatant samples, along with standard solutions containing PUT, SPD, and SPM, were subjected to dansylation treatment as per the method outlined by Scaramagli et al. [25]. The resulting dansylated derivatives were extracted using toluene and were dried and reconstituted in acetonitrile. The separation and quantification of polyamines (PAs) were conducted using high-performance liquid chromatography (HPLC) coupled with fluorescent detection (Nexera XR, Shimadzu, Kyoto, Japan) employing a reverse-phase C18 column (Spherisorb ODS, 2.5-μm particle diameter, 4.6 × 250 mm, Waters, Wexford, Ireland). The following five steps of the programmed acetonitrile:water (*v*/*v*) gradient were applied: 60 to 70% of acetonitrile in 5.5 min, 70 to 80% in 1.5 min, 80 to 100% in 2 min, 100 to 100% in 2 min, 100 to 70% in 2 min, and 70 to 60% in 2 min at a flow rate of 1.5 mL/min. Eluted compounds were identified using a prominence fluorescence detector RF-20A (365 nm excitation, 510 nm emission). A post-run program Lab Solutions was used to integrate the areas of the peaks originated from dansylated polyamines. The results of PA quantification were expressed as mg per kg of dry weight (mg/kg d.w.).

### 2.3. Carbon and Nitrogen Elemental Analysis

Tiny tin capsules, each containing approximately 25 mg of each fungal sample in powdered form, were incinerated in the combustion box of an elemental analyzer (model Elementar Vario EL III, Hanau, Germany) for approximately 2 min at 900 °C. During this process, carbon and nitrogen from the samples were transformed into CO_2_ and N_2_, respectively. Analytical columns were employed to separate the resulting gasses, and their quantification relative to the standard was achieved through variations in the thermal conductivity detector (TCD). Acetanilide (with a known C content of 71.09% and an N content of 10.36%) served as the standard, and the percentages of carbon and nitrogen in the samples were automatically computed according to the method of Karthikeyan and Kumaravel [26]. Furthermore, the total protein content (TP) was determined by calculating it from the N content.

### 2.4. Measurement of Total Phenolic and Total Flavonoid Contents and Assessment of Antioxidant Capacity

#### 2.4.1. Extract Preparation

All experiments were conducted using 80% methanol (MeOH) extracts of fungal species. To prepare the extracts, 50 mg of powdered samples from dried basidioms were macerated with 2 mL of 80% MeOH using a magnetic stirrer (MS-3000 High speed magnetic stirrer, Biosan, Riga, Latvia) for 20 min. The mixture was then subjected to centrifugation at 4 °C for 20 min (Microcentrifuge 5424 R, Eppendorf, Germany). The resulting extracts were further analyzed, and the filtrate was stored at +4 °C, reaching a final concentration of 25 mg/mL.

#### 2.4.2. The Total Phenolic Content

The total phenolic content (TPC) was determined using the Folin–Ciocalteu (FC) reagent following the method outlined by Singleton et al. [27]. In brief, 30 µL of the MeOH extract or a standard (gallic acid, 0.625–80.0 µg/mL) was combined with 150 µL of a 0.1 M FC reagent, followed by a 10 min incubation. Subsequently, 120 µL of 0.7 M sodium carbonate was added, and absorbance was measured after 2 h at 720 nm (Spectrophotometer—MultiScan Go, ThermoScientific, Waltham, MA, USA). All tests were conducted in triplicate, and the TPC was expressed as milligrams of gallic acid equivalents per gram of dry weight (mg GAE/g d.w.).

#### 2.4.3. The Total Flavonoid Content

The total flavonoid content (TFC) was carried out through a colorimetric method, as described by Chang et al. [28]. In summary, 30 µL of the MeOH extract or a standard (quercetin, 0.625–80.0 µg/mL) was mixed with 6 µL of 10% aluminum chloride, 6 µL of 1 M sodium acetate, 90 µL of methanol, and 170 µL of distilled water. Absorbance was measured after 30 min at 415 nm. The tests were performed in triplicate, and the results were expressed as milligrams of quercetin equivalents per gram of dry weight (mg QE/g d.w.).

#### 2.4.4. Antioxidant Capacity

DPPH assay: The ability of the extracts to neutralize the 2,2-diphenyl-1-picrylhydrazyl (DPPH) radical was assessed following the procedure outlined by Espín et al. [29] with slight modifications. In summary, 10 µL of the sample was combined with 100 µL of a 90 µM DPPH solution in MeOH and 190 µL of MeOH. The absorbance was recorded following a 30 min incubation period in a dark place at 515 nm.ABTS assay: This assay involves spectrophotometric monitoring of the conversion of the blue-green colored cation radical ABTS^•^^+^ into its neutral, colorless form and was conducted following the method described by Arnao et al. [30]. ABTS^•^^+^ was generated by directly reacting a 7 mM ABTS solution with 2.45 mM of K_2_S_2_O_8_. Subsequently, 10 µL of the fungal extracts was added to 290 µL of an ABTS solution and was mixed. The absorbance of the sample was read at 734 nm after 5 min of incubation at room temperature.NO assay: The inhibition of the nitric oxide radical (NO^•^) was evaluated using the Griess diazotization process, as outlined in the methodology developed by Green et al. [31]. The reaction mixture consisted of 15 µL of the extract, 250 µL of 10 mmol/L of sodium nitroprusside and 250 µL of a phosphate buffer (pH 7.4). After incubation for 90 min at room temperature under constant light, 500 µL of a Griess reagent (a combination of a 0.2% solution of N-(1-naphthyl)-ethylenediamine dihydrochloride and a 2% solution of sulfanilamide in 4% phosphoric acid) was added. The degree of inhibition was gauged by quantifying the absorbance of the resultant chromophore at 546 nm.FRAP assay: This assay was carried out according to Benzie and Strain [32]. The freshly prepared FRAP (Ferric Reducing Antioxidant Power) reagent comprises a solution containing 10 mmol/L of TPTZ in 40 mmol/L of HCl, 0.02 mmol/L of FeCl_3_ × 6H_2_O, and an acetate buffer (pH 3.6) in a ratio of 10:1:1. Briefly, 10 µL of each extract was combined with 225 µL of the FRAP reagent and 22.5 µL of distilled water (dH_2_O). Absorbance was measured after 6 min at 593 nm.

The radical scavenging capacity (RSC) against DPPH, ABTS, and NO radicals, as well as the reduction potential in the FRAP assay, were evaluated by establishing a standard curve of Trolox. The outcomes were quantified and presented as millimoles of Trolox equivalents per gram of dry fungal material, either fresh or dry weight (mmol TE/g d.w.), depending on the particular extract utilized in the assay.

### 2.5. Neuroprotective Activity

The neuroprotective activity of the extracts was evaluated by assessing their ability to inhibit the AChE enzyme using Ellman’s method [33]. In brief, 20 µL of AChE (0.5 U/mL) was combined with 110 µL of a 20 mM Tris-HCl buffer at pH 8, along with 10 µL of the extract. In the blank sample, AChE was substituted with the 20 mM Tris-HCl buffer at pH 7.5, and in the control, the 20 mM Tris-HCl buffer at pH 8 was added instead of the sample. The 96-well plate was then incubated for 15 min at 37 °C with continuous shaking. Following incubation, 40 µL of a 3 mM solution of 5,5′-dithio-bis(2-nitrobenzoic acid) and 20 µL of 15 mM acetylthiocholine iodide were added to the plate. The absorbance at 412 nm with a total of 15 measurements was taken at 1 min intervals. The percentage (%) of enzyme inhibition was calculated using the equation detailed in Mišković et al. [3]. The reference inhibition time was set at 10 min or 600 s, respectively.

### 2.6. Statistical Analysis

This study employed a variety of statistical methods, encompassing descriptive statistics, a one-way Analysis of Variance (ANOVA) with Tukey post hoc tests, a Principal Component Analysis (PCA), dendrogram hierarchical clustering, as well as a Pearson correlation analysis. In the context of one-way ANOVA, the research investigated the differentiation among analyzed fungi species, assessing the statistical significance of these distinctions using the Fisher (F) test and presenting the results as “*p*-values”. All statistical data processing was conducted using the R programming environment. Descriptive statistics, two-way ANOVA, and *t*-tests were executed with the “rstatix” R package [34]. Dendrogram clustering utilized the “dendextend” R package [35], while diverse data visualizations were generated with the “ggplot2” R package [36].

## 3. Results and Discussion

### 3.1. Mycochemical Analysis

The mycochemical analysis of eight fungal species included the assessment of PA content, TP, TPC, and TFC. For the majority of the analyzed species, with the exception of the *Ganoderma* species, there is only a limited amount of available data, making it challenging to compare results due to differences in the methodological approach.

#### 3.1.1. Polyamine Content

The results of the HPLC analysis of the selected PAs—PUT, SPD, and SPM—in eight fungal species are presented in Figure 1 and represent the first report of PA content in these species.

PUT and SPD were quantified in all analyzed samples, while SPM was not detected in *L. nuda*. It is important to note that the predominant PA was SPD in a range from 3.86 to 71.58 mg/kg d.w. (Figure 1). The highest contents of PUT and SPD were observed in *C. odora* (63.30 ± 1.60 and 64.64 ± 0.29 mg/kg d.w., respectively) and *P. caesia* (55.87 ± 3.16 and 62.98 ± 3.19 mg/kg d.w., respectively), while *C. aegerita* contained the highest amount of SPD (71.58 ± 6.47 mg/kg d.w.). On the other hand, SPM was predominant in *M. elata* (1.49 ± 0.14 mg/kg d.w.), followed by *C. aegerita* (0.75 ± 0.42 mg/kg d.w.). The highest content of total polyamines in the inspected fungi was recorded in the following order: *C. odora > P. caesia > C. aegerita > M. elata > L. nuda > G. applanatum* > *C. prunulus* > *G. resinaceum*. In contrast to plants, the significance of PA content in fungi is still not fully recognized, although it is known that PAs affect fungal growth and development through the entire fungal life cycle including the sporulation, growth, development, and modulation of host interactions [14]. To the best of our knowledge, PA content was not determined for species investigated in this study. However, the quantification of PAs is mostly conducted in cultivated species, including *Agaricus bisporus, Lentinula edodes*, and *Pleurotus spp.* For example, Dadakova et al. [24] summarized the PA content in thirteen cultivated mushrooms, where the most prevalent PA was SPD (22.2–195.0 mg/kg wet weight and 157–522 mg/kg d.w., respectively), followed by PUT (0.9–90.6 mg/kg w.w. and 3.8–59.4 mg/kg d.w.), while SPM was not detected in all species, which is in accordance with the results of this study. A decade later, Reis et al. [6] quantified only SPD in fresh, cooked, and canned mushroom *A. bisporus* (6.4–8.5 mg/100 g). On the other hand, there is scarce information regarding the PA content in wild-growing mushrooms [12,24]. Dadakova et al. [24] analyzed the content of six biogenic amines, including three Pas in seventeen wild-growing edible mushrooms from Poland, where *Boletus erythropus* contained the highest amount of SPD (2219 mg/kg d.w.). For instance, wild-growing mushrooms in Turkey have been documented as significant sources of SPD, while in *Xerocomus badius*, *X. chrysentereon*, and *Suillus variegatus*, PUT was the most prevailing amine [12,24].

#### 3.1.2. Total Protein Content

Table 2, among others, presents the results of TP, while the results of the CHN analysis are provided in Appendix A. The TP content, calculated based on nitrogen (N) content, was the highest in *L. nuda*, *C. prunulus*, and *C. odora* (47.40, 46.70, and 39.70%, respectively). The TP content in *C. odora* was twice as high as that found in *C. odora* from Portugal (TP = 17.33 ± 1.37%) according to Vaz et al. [37], which was also the case with *G. resinaceum* from Serbia as well where TP was 4.7% [38]. A similar trend was noted in *C. prunulus*, *C. aegerita*, and *L. nuda*, where the species from other studies exhibited more than a 50% lower TP content (18.13 ± 0.37%, 6.68 ± 0.26%, and 8.11–12.18%, respectively) [39,40,41,42], while the TP content of *L. nuda* from West Macedonia and Epirus was in accordance with our results [43].

#### 3.1.3. Total Phenolic and Total Flavonoid Contents

*C. prunulus* and *G. applanatum* exhibited the highest TPC values (49.02 ± 0.59 and 45.75 ± 5.36 mg GAE/g d.w., respectively), followed by *M. elata, L. nuda*, and *C. aegerita* (Table 2). *G. applanatum* also displayed the highest TFC content (7.02 ± 0.89 mg QE/g d.w.). In contrast, *P. caesia* had the lowest TPC, with no detectable TFC. The TPC of commercial and wild samples from pine and oak forests of *L. nuda* MeOH extracts from Portugal was significantly lower compared to the results of this study (15.98–20.54 mg GAE/g d.w.) [40]. The obtained TPC in a *C. aegerita* extract was also greater compared to other samples from Serbia, where EtOH (17.36 ± 0.88 mg GAE/g d.w.) and MeOH (13.80 ± 0.21 mg GAE/g d.w.) extracts were analyzed [41,44]. The most impressive difference in TPC was observed in *M. elata*, since the MeOH extract from Australia exhibited 86 times lower content (0.46 ± 0.01 mg GAE/g d.w.) [45]. Furthermore, it is noteworthy that the TFC of the *C. aegerita* MeOH extract from other parts of Serbia [44] was quantified at a very low level (0.73 ± 0.37 mg QE/g d.w.), contrary to the findings in this study. Conversely, the TPC and TFC in MeOH extracts of *C. odora* from Turkey [46] were significantly higher (TP: 82.646 ± 1.623 mg/g; TFC: 117.753 ± 3.491 mg/g) compared to the EtOH extract from Serbia (38.112 ± 0.251 mg/g) [47]. On the other hand, the TPC and TFC content of the *Ganoderma* species are in accordance with the analyzed MeOH extracts from Turkey [48], while compared to our previously published data, EtOH extracts had a higher content of phenolics and flavonoids [8,9,49]. All of this further indicates that the mycochemical composition of fruiting bodies is significantly influenced by their growth habitat and the type of extraction solvent polarity [3,40]. The mycelia of *G. applanatum* from Poland contained a significantly lower TPC in contrast to *G. resinaceum*, where the TPC was in accordance with our results [22]. It is important to highlight that this study represents the first report of TFC content in *C. prunulus, L. nuda*, and *M. elata*.

### 3.2. Antioxidant Potential

All tested fungal species exhibited antioxidant activity, as shown in Table 2. The most notable variations in the strength of antioxidant activity among the analyzed species were observed in the ABTS, FRAP, and DPPH assays. Conversely, no differences were detected in the NO assay. However, it is important to note that the activity of extracts in this assay showed a significant correlation with both primary and secondary metabolites (Figure 2).

Conversely, PAs did not correlate with the analyzed antioxidant activity. The highest correlation was observed between the reduction potential (FRAP assay) and TPC, while the scavenging potential for the DPPH and NO radicals demonstrated significant correlations both with the TPC and TP.

The observed antioxidant activities among the tested fungal species reveal notable variations, with *G. applanatum* and *L. nuda* emerging as the most potent contributors to antioxidative potential. This outcome is attributed to their high scavenging ability and impressive reduction potential, as outlined in Table 2. The strongest ability to capture the DPPH radical was observed in the following order: *L. nuda* > *C. odora* > *C. prunulus* > *M. elata* > *G. applanatum* (Table 2). *G. applanatum, G. resinaceum, M. elata*, and *L. nuda* showed the highest neutralization of the ABTS radical (70.42 ± 2.60, 69.70 ± 2.54, 68.88 ± 1.13, and 60.93 ± 4.72 mg TE/g d.w., respectively), while the most potent reduction potential was demonstrated by *G. applanatum* and *L. nuda* (19.24 ± 1.77 and 18.32 ± 2.89 mg TE/g d.w., respectively). In contrast, the lowest reduction potential, together with the DPPH and ABTS scavenging ability, was observed in *P. caesia* and *C. aegerita* (Table 2). Also, it is important to note that *G. resinaceum* showed low antioxidant activity, with the exception of the ABTS radical neutralization (69.70 ± 2.54 mg TE/g d.w.).

Notably, the antioxidant activities of the *Ganoderma* species observed in our study align with several previous investigations. For instance, Zengin et al. [48] reported a higher DPPH and FRAP potential in MeOH extracts of *G. applanatum* and *G. resinaceum* from Turkey, attributing this to its rich content of phenolic compounds, while the ABTS scavenging ability was weaker compared to our results (14.85 ± 1.31 and 41.32 ± 0.39 mg TE/g extract, respectively). The same trend is observed when comparing the results of this study to the antioxidant study of these two *Ganoderma* species from Poland, where MeOH was also used as the extraction solvent [22]. Similarly, the robust antioxidative performance of *G. applanatum* from India and Kenya, especially in DPPH radical scavenging and reduction potential, has been corroborated by studies conducted by Rajoriya et al. [50] and Siangu et al. [51]. On the contrary, the cultivated mycelia of *G. applanatum* extracted with EtOH had a weaker DPPH activity (20.35% at the concentration of 20 mg/mL), compared to the scavenging ability observed here [52]. Nevertheless, in our previous investigations, EtOH extracts obtained from the *Ganoderma* species native to Serbia exhibited a markedly greater antioxidant capacity [8,9,49]. A similar pattern was noted in *G. applanatum* sourced from China, wherein EtOH extracts demonstrated notable antioxidant activity, achieving 91.76% DPPH inhibition (IC_50_ = 0.05642 mg/mL) and 100% ABTS activity (IC_50_ = 0.01962 mg/mL) [53]. This variance could be ascribed to differences in solvent polarity and variations in the experimental protocols employed, since it is well known that solvent polarity affects phenolic content and, consequently, impacts antioxidant activity [54].

Our study reveals the significant antioxidant potential in *L. nuda* MeOH extracts, but the lack of extensive comparative data on *L. nuda*’s antioxidant activity makes it challenging to contextualize our findings within the broader scientific landscape. Pinto et al. [40] compared the antioxidant activity of the commercial samples of a fruiting body and the in vitro cultured mycelia of *L. nuda* with wild samples of a fruiting body. Their results revealed that the iMMN culture medium allowed for the highest antioxidant potential, followed by the commercial fruiting bodies, while the wild types showed a lower activity and TPC [40]. When compared to our results, it can be observed that the DPPH activity of the wild type of this fungal species from Portugal was higher (EC_50_ = 15.48 ± 0.23 mg/mL) [40], while the EtOH extract of *L. nuda* from Turkey showed a lower DPPH activity with a range of 2.79% to 50.20% inhibition [55]. The observed antioxidant activity suggests that *L. nuda* may hold promise as a natural source of antioxidants, but caution is warranted in drawing definitive conclusions without a more extensive comparative framework.

*C. odora* demonstrated a pronounced neutralization of the DPPH radicals, showcasing notable antioxidant activity, comparable to the results of the MeOH extracts from Turkey, where a high DPPH scavenging ability was noticed (73.38 ± 1.60% at the concentration of 2 mg/mL) [46]. Moreover, Vaz et al. [37] performed two different extraction methods to obtain extracts of *C. odora* from Portugal with high molecular weight compounds, such as polysaccharides, and low molecular weight compounds, such as phenolic compounds. Their results demonstrated a twice-as-high DPPH activity of a water-soluble polysaccharide extract (EC_50_ = 3.56 ± 0.13 mg/mL) compared to the tested EtOH extract (EC_50_ = 6.77 ± 0.05 mg/mL), which was in accordance with the extraction yield, since EtOH fractions were lower compared to water soluble ones [37]. In contrast, the EtOH extract from a wild sample gathered in the Niš region of Serbia displayed comparatively lower antioxidant activity, likely attributed to the reduced levels of antioxidant components present in the extract [47]. This further indicates the importance of fungal geographical origin and the polarity of the solvent for the extraction yield of bioactive compounds and, consequently, the investigated bioactivity.

*C. prunulus* and *M. elata* also demonstrated notable antioxidant activity, and, to the best of our knowledge, this is the first report on the antioxidant activity of these species from Serbia and the Balkan region in general. In the literature data, there is only one available study of *C. prunulus* from Portugal, where a moderate DPPH scavenging ability was observed, owing to the lower content of TPC, while the highest level of ascorbic acid was detected [39]. However, compared to our results, a sample from Portugal exhibited higher DPPH activity and reduction potential with EC_50_ values of 1.75 ± 0.13 mg/mL and 3.36 ± 0.03 mg/mL, respectively [39]. Also, the ethyl acetate and MeOH extracts of *M. elata* from India showed a higher DPPH, NO, and ABTS neutralization compared to the results of this study [56], while Kalyoncu et al. [57] reported a lower antioxidant activity of *M. elata* from Turkey, with 59.22% of DPPH inhibition. The reduction potential of this species collected in Australia was also significantly higher (63.0 ± 0.3 mmol Fe[II]-E/g extract) [45], supporting the notion that habitat ecology, the extraction procedure, and solvent polarity may affect antioxidant activity.

Conversely, the comparatively lower antioxidant activity observed in the *C. aegerita* in our study is consistent with the findings of Karaman et al. [44], where a low DPPH activity and reduction potential were detected (45.3 ± 0.35 mg TE/g d.w. and 10.74 ± 0.09 mg TE/g d.w., respectively), while Petrović et al. [41] reported a higher DPPH activity of an MeOH extract (EC_50_ = 7.23 ± 0.18 mg/mL). Also, higher DPPH and NO scavenging abilities were demonstrated for the EtOH extract of *C. aegerita* from India at the same concentration tested—20 mg/mL (85.63 ± 0.12% and 82.02 ± 0.12%, respectively) [58]. Concerning the *P. caesia* extract, the notably low antioxidant activity may be linked to its very low TPC. It is crucial to underscore the pioneering nature of this research, as there is currently no existing literature data regarding the antioxidant activity of this particular species. Hence, the investigation into the antioxidant activity of *P. caesia* has provided valuable insights, although the scarcity of comparable literature on this specific mushroom presents both a challenge and an opportunity for further research.

All these observations align with the concept that the antioxidant capabilities can vary significantly across distinct fungal taxa.

### 3.3. AChE Inhibitory Potential

Among the eight different fungal extracts, only three did not exhibit anti-AChE activity (Figure 3).

The strongest potency for the inhibition of the AChE enzyme was observed for *L. nuda* (99.94 ± 3.10%)*, G. applanatum* (98.05 ± 0.83%), and *P. caesia* (88.21 ± 4.76%), which is higher compared to the percent of inhibition of the positive control (donepezil) (87.44%). *M. elata* and *C. prunulus* exhibited a moderate level of anti-AChE activity, whereas *C. odora, C. aegerita*, and *G. resinaceum* demonstrated an inhibition exceeding 100% across the tested concentration range (Figure 3).

The observed high anti-AChE activity of *L. nuda, G. applanatum*, and *P. caesia* suggests that these species may contain bioactive compounds capable of interfering with the AchE enzyme, which is implicated in neurodegenerative disorders such as Alzheimer’s disease. Moreover, various species of *Ganoderma*, especially *G. lucidum*, have been reported to possess neuroprotective properties and exhibit cholinesterase inhibitory activity [8,22,59]. Tel-Cayan et al. [60] documented anti-AChE activity in four distinct types of extracts from *G. adspersum* sourced from Turkey, with the MeOH extract demonstrating an inhibition rate of 41.34 ± 3.79, which was more than two times lower in comparison with this study for *G. applanatum*. Moreover, Kozarski et al. [38] reported that the hot-water extract of *G. resinaceum* at a concentration of 1 mg/mL achieved 81.6 ± 6.5% anti-AChE activity, which was significantly higher than the results from our study. At the same time, Rašeta et al. [8] observed comparable anti-AChE effects in water extracts from four distinct autochthonous *Ganoderma* species from Serbia (*G. applanatum*, *G. lucidum*, *G. pfeifferi*, and *G. resinaceum*). The reported activity fell within the 1.04–1.05 mg GALAE/g extract range. In a subsequent study conducted two years later, Sułkowska-Ziaja et al. [22] documented comparable activity in MeOH extracts derived from the mycelial cultures of four specific *Ganoderma* species, *G. adspersum*, *G. applanatum*, *G. carnosum*, *G. lucidum*, *G. pfeifferi*, and *G. resinaceum*, with reported values in the range of 1.19 to 1.22 mg GALAE/g extract. However, extracts from *G. applanatum* and *G. resinaceum* did not exhibit any discernible activity.

Additionally, Akata and colleagues [61] documented anti-AchE effects in various fungal species, including *Agaricus campestris*, *Coprinus comatus*, *Leucoagaricus leucothites*, *Lycoperdon utriforme*, *Macrolepiota mastoidea*, and *Macrolepiota procera*, sourced from diverse regions in Turkey. They observed that the activity was notably lower (ranging from 0.83 to 0.97 mg GALAE/g extract) compared to our previously reported study involving *Ganoderma* species using the same experimental procedure [8].

The concentration of phenolic compounds from *Ganoderma* and other fungal species was attributed to AChE inhibition by many authors [62,63,64,65,66]. Hence, differences in the inhibition of AChE observed in our study between the two *Ganoderma* species may be linked to variations in their secondary metabolite profiles, since the *G. applanatum* extract demonstrated twice the TPC compared to the extract of *G. resinaceum*, which did not exhibit anti-AChE activity at the tested concentration. Noteworthy is that *G. applanatum* contained a higher TP compared to the extract of *G. resinaceum*, which may have had an effect on the obtained activity. In our previously published study, it was observed that *G. applanatum* EtOH extracts exhibited a fivefold higher TPC compared to the corresponding extract from *G. resinaceum* (265.38 ± 0.81 and 50.87 ± 0.29 mg GAE/g d.w., respectively). Interestingly, despite this difference in TPC, both extracts demonstrated equivalent levels of activity [8].

For instance, polysaccharides isolated from both the fruiting bodies and primordia of *G. lucidum* demonstrated neuroprotective properties [21,54,59,67]. In addition, based on the study conducted by Liu et al. [68], it is known that polysaccharides from *C. aegerita* significantly prolong the lifespan of *Drosophila melanogaster* as a model organism and alleviate the oxidative stress induced by H_2_O_2_. Interestingly, *G. applanatum*, which exhibited significant AChE inhibition, also demonstrated a higher concentration of PUT and SPD compared to *G. resinaceum*. The presence of the elevated levels of PUT and SPD in *G. applanatum* may also contribute to its pronounced AChE inhibitory effects, since these PAs have been implicated in neuroprotective mechanisms and may interact with AChE, potentially altering its catalytic activity due to its positive charge and chaperone-like activity [15,69]. This is supported by the results of the correlation analysis, which showed a high positive correlation between AChE inhibition and levels of PUT and SPD (Figure 3). Conversely, *G. resinaceum*, which did not show a notable AChE inhibition in our study, displayed a distinct PA profile. The lower levels of specific PAs in *G. resinaceum* may suggest a reduced impact on AChE activity. This correlation raises intriguing questions about the connection between PAs and AChE inhibition in the *Ganoderma* species and suggests a possible synergistic effect.

In this study, the investigation into AChE inhibition among other investigated fungal species marks a novel exploration in the field. To the best of our knowledge, prior to this research, the AChE activity in *L. nuda*, *P. caesia*, *C. aegerita*, *C. odora*, *M. elata*, and *C. prunulus* has remained largely unexplored. Our findings reveal a previously unrecognized potential for AChE modulation within these fungal species.

Polysaccharides isolated from *L. nuda* showed antioxidant and immuno-modulatory activities [70], while the substantial anti-AChE activity observed in the *L. nuda* extract may be attributed to a synergistic influence of both secondary and primary metabolites. This is supported by the presence of a high TPC, moderate levels of PUT and SPD, and a high TP content quantified in this extract. On the other hand, a high inhibition of the AChE enzyme by the extract of *P. caesia* coincides with an elevation in specific PAs, including PUT and SPD, while the TPC and TP were among the lowest. This supports the indications that PAs may play a crucial role as neuroprotective agents [69,71,72,73].

The anticipated inhibitory activity against the target enzyme was not expressed in *C. odora* and *C. aegerita*, despite the high PA levels. This suggests the possibility that the assessed concentration might fall below the effective threshold required for the manifestation of the desired biological activity and that the selected concentration may not have been optimal for eliciting the anticipated responses. Hence, a broader concentration range or alternative formulations may be necessary to uncover the latent pharmacological potential of these fungal species.

The variations in anti-AChE activity across different mushroom species, as evidenced by the contrast within this study, underscore the importance of species-specific considerations. Differences in phytochemical profiles, environmental factors, and genetic variations among mushrooms may influence their bioactivity [3]. This emphasizes the significance of comprehensive studies to understand the complex effects of individual fungal species on AChE inhibition. Furthermore, the connection between PAs and neuroprotective activity has not been completely understood, although it has been suggested that it is closely linked to PA machinery. Especially, the SPM and SPD accumulated in glia brain cells, due to their multiple positive charges, affect many neuronal and glial receptors, channels, and transporters and, therefore, affect many neurologic diseases including global amnesia, depression, stress, anxiety, autism, glioblastoma multiforme, glaucoma, migraines, neuropathic pain, sleeplessness, and drug addiction [74]. Furthermore, PAs are prone to the cause-forming of protein aggregation and to accelerating the rate of fibrillization in lysozymes, which can lead to neuropathic and non-neuropathic amyloidosis, which are associated with Alzheimer’s and Parkinson’s disease as well as with aging [75]. Intriguingly, it has been documented that PAs improve social memory formation, synoptic plasticity, behavior, and learning but also increase the production of Pas, as has been reported for Alzheimer’s disease, although biphasic behavior and ambiguous effects PAs to AChE activity has been established [76]. Although PAs are generally proposed to support cholinergic activity, according to Kossorotow et al. [77], at a low micromolar concentration range, SPM and SPD may either stimulate or inhibit AChE activity depending on the amounts of acetylcholine, whereby at low acetylcholine amounts, an inhibitory effect of PAs on AChE prevails. Moreover, it has been documented that PAs are powerful regulators of Alzheimer’s disease through dietary interventions and microbiota manipulations, such as probiotic supplementation, e.g., *Bifidobacterium animalis subsp. lactis* LKM512 [78]. This supplementation of gut probiotics resulted in an increase in PA levels in the intestines and alleviated inflammation in the colon [79], while on the other hand, it affected the brain’s memory and mice’s social behavior during AD [78]. A potential avenue for future research in this area could involve elucidating the effects of a PA-abundant fungi diet on mice’s PA metabolism and the development of AD.

Our study sheds light on this uncharted territory, as we demonstrate, for the first time, that PAs in these fungal strains may exhibit neuroprotective properties. This discovery challenges the current understanding of the role of PAs in fungal biology and underscores the need for future investigations into the potential therapeutic applications of fungal PAs in neuroprotection. However, it is crucial to acknowledge the limitations of our study and the need for further investigations to elucidate the specific compounds responsible for the observed inhibitory effects. The identification of these compounds could provide insights into the underlying mechanisms and could contribute to the development of targeted therapies for neurodegenerative conditions. Moreover, in vivo studies are crucial to validate our in vitro findings and to assess the potential bioavailability and pharmacokinetics of active compounds from edible fungi.

### 3.4. PCA Analysis

The PCA was employed to discern patterns and relationships among the quantified compounds (TP, TPC, TF, and PAs) and the antioxidant and anti-AChE activity of all investigated fungal species (Figure 4).

The first two principal components, PC1 and PC2, accounted for 67.23% of the total variance, indicating a substantial representation of the dataset. The results of the PCA highlight the inherent variability in our dataset and offer insights into the key factors influencing the observed trends. A stronger separation was conducted in the horizontal plane of the PC2, with a TP and TPC loading in the I quadrant and PAs in the II quadrant.

Significantly, both PUT and SPD demonstrated a high positive loading on both PCs, along with anti-AChE, indicating their notable involvement in this activity—a correlation that is further supported by the correlation analysis results. Additionally, these variables separate *C. aegerita, C. odora*, and *P. caesia* from the other fungal species, suggesting that PAs play a pivotal role in influencing the observed AChE-inhibitory activity of the selected fungi. This is supported with the results of PA quantification, since the highest levels of SPD and PUT were detected in these species. While *C. aegerita* and *C. odora* did not demonstrate AchE-enzyme inhibition, the levels of PUT and SPD in *P. caesia* likely exerted a significant influence on this activity. This is evident as *P. caesia* exhibited one of the most robust anti-AChE activities among the tested fungal species. When it comes to the SPM levels, the right angle with the anti-AChE vector indicates that there is no correlation between these two variables.

Conversely, TP and TPC displayed a negative loading on PC2 together with variables of antioxidant assays, indicating its inverse relationship with the anti-AChE activity and PA levels. Within the first quadrant, TPC showed a notably strong positive correlation with the reduction potential (FRAP assay) and a slightly weaker positive correlation with the NO and DPPH scavenging ability. Conversely, DPPH exhibited a strong correlation with TP, followed by the NO and FRAP activity. These correlations align with those observed in the heatmap from the correlation analysis (Figure 2). In the negative sector of both PC1 and PC2, TFC and ABTS were present, indicating a robust positive correlation between these variables, as confirmed by the correlation analysis as well. Nevertheless, the clustering of the *Ganoderma* species and *M. elata* in the negative part of PC is in accordance with the results of this study, since these species had the strongest ABTS scavenging ability and the highest levels of TFC. On the other hand, the clustering of *C. prunulus* and *L. nuda* are characterized with a high TPC and TP and a strong DPPH, NO, and FRAP activity and lower PA levels, which is in accordance with the results obtained.

The clustering observed in the score plot hints at potential subgroups which exhibit different levels of tested activities and active compounds, but that grouping is not based on the tropism nor the taxonomic placement of the investigated species, except for the closely related *Ganoderma* species. Hence, we assume that different environmental factors and species-specific characteristics affect variations in the detected compounds and activities, prompting a further investigation into the biological factors contributing to these distinct patterns.

## 4. Conclusions

In conclusion, the findings presented in this study highlight the neuroprotective and antioxidant properties exhibited by PAs and phenolic compounds extracted from selected edible and medicinal fungi. Furthermore, this investigation represents the first documentation of PA levels in the inspected fungi species. It also signifies the initial exploration of the neuroprotective capabilities of *P. caesia*, *C. odora*, *C. prunulus*, and *M. elata*, along with the pioneering examination of the antioxidant potential of the *P. caesia* extract. The species with the most potent anti-AchE and antioxidant activities were *G. applanatum* and *L. nuda*, which was attributed to the synergistic effects of TP, TPC, and PAs. Additionally, *P. caesia* demonstrated significant AChE inhibition, primarily associated with elevated SPD and PUT levels, while the significant antioxidant potential of *C. odora* was related to the synergism of primary (TP) and secondary metabolites (TPC). Besides the experimental results, a correlation analysis and PCA also confirmed the connection of PA levels in the selected fungal species with AChE inhibition, while TPC was closely correlated with the antioxidant activities estimated by the DPPH, ABTS, and NO assays.

Moreover, the different fungal species investigated in this study revealed intriguing variations in the composition and efficacy of PAs and phenolic compounds and emphasize the potential for harnessing fungal resources in the development of therapeutic strategies for neurological disorders as well as through dietary interventions.

## Figures and Tables

**Figure 1 jof-10-00021-f001:**
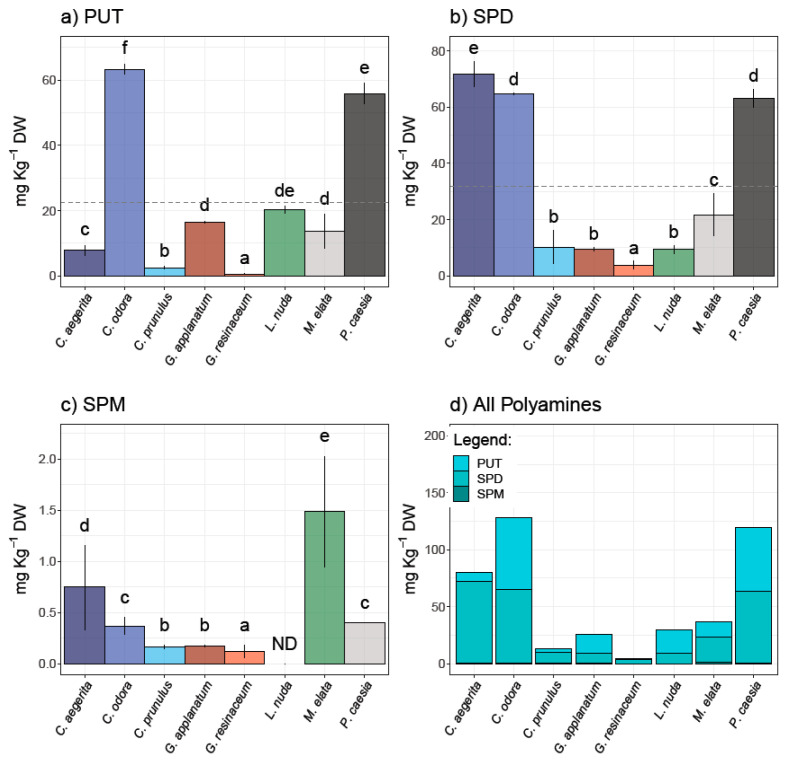
Polyamine content in analyzed fungal species—(**a**) putrescine (PUT), (**b**) spermidine (SPD), and (**c**) spermine (SPM)—as well as (**d**) total PAs. Different small letters indicate significant differences among different species; Tukey’s significant difference (HSD) post hoc test (*p* ≤ 0.05). Data represent the mean ± standard deviation (SD).

**Figure 2 jof-10-00021-f002:**
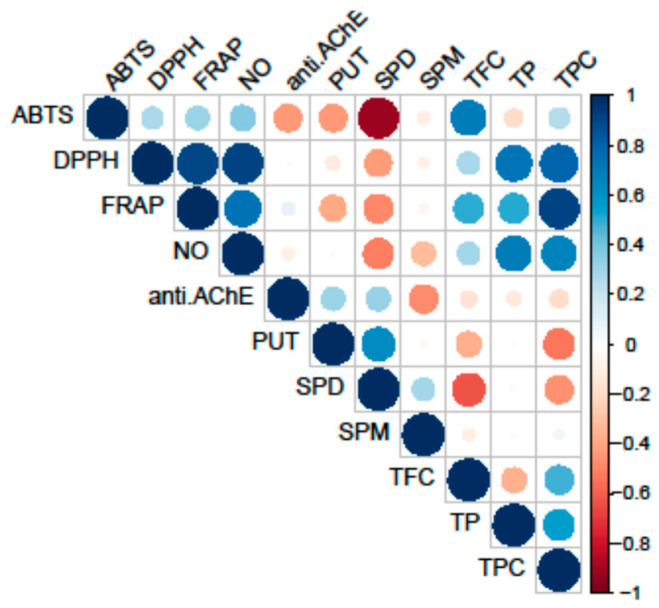
Pearson’s coefficient of the correlation matrix of the examined parameters in eight selected fungal species collected on the territory of the Republic of Serbia. Blue squares represent a highly significant correlation of inspected parameters, while red squares present low interactions, assessed according to the corresponding Pearson’s coefficient. The following are the abbreviations of the examined parameters: ABTS—radical scavenger capacity against 2,2′-azinobis(3-ethylbenzothiozoline)-6-sulfonic acid, ABTS^•+^; DPPH—radical scavenger capacity against 2,2-diphenyl-1-picrylhydrazyl radical, DPPH^•^; FRAP—ferric-reducing antioxidant power; NO—radical scavenger capacity against NO radical; anti.AChE—anti-acetylcholinesterase activity; PUT—putrescine; SPD—spermidine; SPM—spermine; TFC—total flavonoid content; TP—total protein content; TPC—total phenolic content.

**Figure 3 jof-10-00021-f003:**
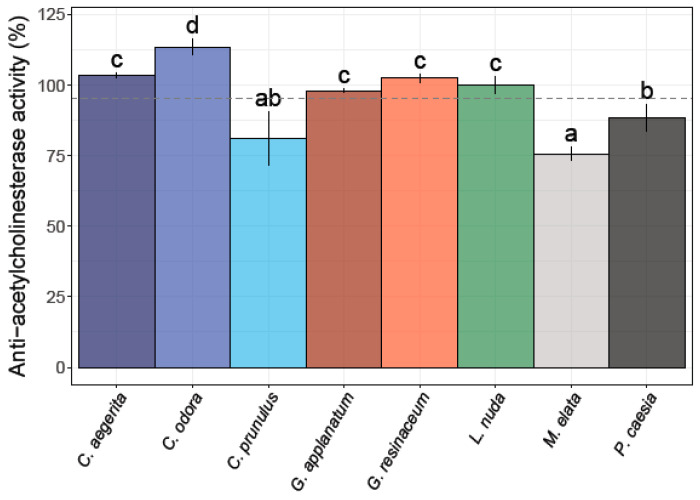
Anti-acetylcholinesterase activity of the eight analyzed fungal species. Different small letters indicate significant differences among different species; Tukey’s significant difference (HSD) post hoc test (*p* ≤ 0.05). Data represent the mean ± standard deviation (SD).

**Figure 4 jof-10-00021-f004:**
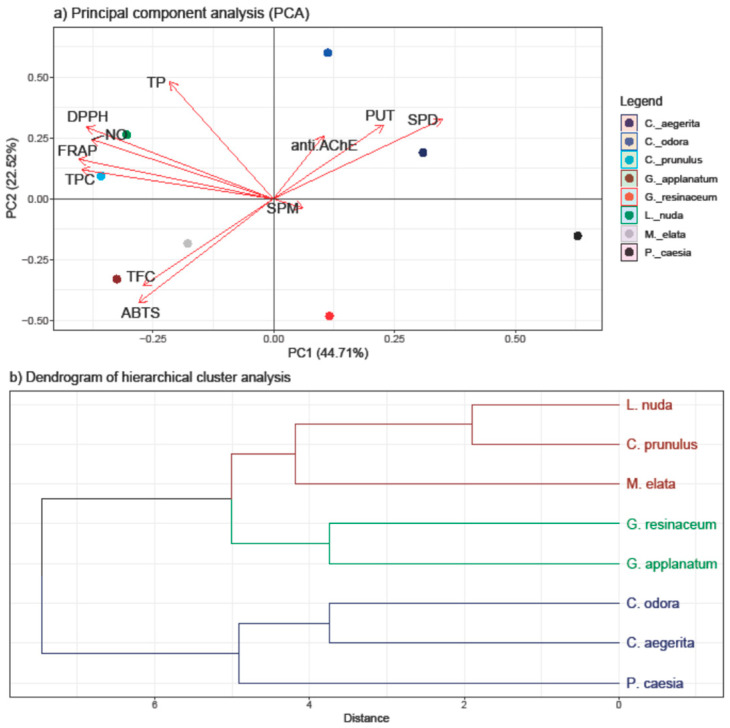
PCA analysis of all examined parameters in the eight selected fungal species collected on the territory of the Republic of Serbia (**a**) and the hierarchical clustering of the eight analyzed fungal species (**b**). The following are the abbreviations of the examined parameters: ABTS—radical scavenger capacity against 2,2′-azinobis(3-ethylbenzothiozoline)-6-sulfonic acid, ABTS^•+^; DPPH—radical scavenger capacity against 2,2-diphenyl-1-picrylhydrazyl radical, DPPH^•^; FRAP—ferric reducing antioxidant power; NO—radical scavenger capacity against NO radical; anti.AChE—anti-acetylcholinesterase activity; PUT—putrescine; SPD—spermidine; SPM—spermine; TFC—total flavonoid content; TP—total protein content; TPC—total phenolic content; Tukey’s honestly significant difference (HSD) post hoc test (*p* ≤ 0.05). Data represent the mean ± standard deviation (SD).

**Table 1 jof-10-00021-t001:** Analyzed fungal species, localities, sampling dates, and voucher numbers.

Species Name	Locality ^1^	Sampling Date	Voucher Number
*Clitocybe odora*	Tara Mountain	29 October 2021	12-00906
*Clitopilus prunulus*	Tara Mountain	29 October 2021	12-00907
*Lepista nuda*	Tara Mountain	29 October 2021	12-01046
*Postia caesia*	Tara Mountain	29 October 2021	12-01047
*Morchella elata*	Petrovaradin Hill	8 April 2023	12-01048
*Cyclocybe aegerita*	Novi Sad town	5 November 2019	12-01049
*Ganoderma applanatum*	Morović’s forest	17 May 2023	12-00714
*Ganoderma resinaceum*	Novi Sad town	21 May 2023	12-00722

^1^ All species were sampled within the territory of the Republic of Serbia.

**Table 2 jof-10-00021-t002:** Mean values with standard deviation and Tukey post hoc test classification of antioxidant profile and total flavonoid (TFC), total protein (TP), and total phenolic content (TPC) among analyzed fungal species.

Species	ABTS	DPPH	FRAP	NO	TFC	TP	TPC
*C. aegerita*	28.08 ± 4.00 ^a^	5.33 ± 0.94 ^b^	14.31 ± 1.42 ^bc^	9.22 ± 1.15 ^a^	ND	30.27 ± 1.24 ^c^	35.72 ± 1.65 ^c^
*C. odora*	39.04 ± 10.88 ^b^	9.01 ± 0.73 ^d^	16.61 ± 2.43 ^c^	10.44 ± 1.19 ^b^	0.21 ± 0.01 ^a^	39.70 ± 0.95 ^d^	29.69 ± 2.04 ^bc^
*C. prunulus*	58.84 ± 5.46 ^d^	8.89 ± 0.78 ^cd^	17.27 ± 1.59 ^cd^	10.80 ± 1.94 ^bc^	1.76 ± 0.01 ^bc^	46.70 ± 1.74 ^e^	49.02 ± 0.59 ^d^
*G. applanatum*	70.42 ± 2.60 ^e^	7.98 ± 0.30 ^c^	19.24 ± 1.77 ^d^	10.41 ± 0.56 ^b^	7.02 ± 0.89 ^d^	17.24 ± 0.49 ^b^	45.75 ± 5.36 ^d^
*G. resinaceum*	69.70 ± 2.54 ^e^	5.16 ± 0.44 ^b^	12.68 ± 1.47 ^b^	9.48 ± 0.56 ^ab^	2.11 ± 0.51 ^c^	13.64 ± 1.09 ^a^	20.20 ± 2.56 ^b^
*L. nuda*	60.93 ± 4.72 ^d^	9.28 ± 0.31 ^de^	18.32 ± 2.89 ^d^	10.96 ± 0.44 ^bc^	1.09 ± 0.00 ^b^	47.40 ± 1.28 ^e^	39.60 ± 3.69 ^c^
*M. elata*	68.88 ± 1.13 ^e^	8.09 ± 0.11 ^c^	16.62 ± 1.06 ^c^	10.10 ± 0.74 ^b^	2.60 ± 0.44 ^c^	34.62 ± 0.89 ^cd^	39.75 ± 2.77 ^c^
*P. caesia*	46.92 ± 7.03 ^c^	3.03 ± 0.29 ^a^	5.40 ± 0.91 ^a^	9.35 ± 0.42 ^a^	ND	18.62 ± 0.42 ^b^	5.56 ± 0.47 ^a^
One-way ANOVA							
*F test*	23.85	50.01	18.08	1.34	26.84	24.88	77.34
*p*	2.48 × 10^−7^	1.06 × 10^−9^	1.71 × 10^−6^	2.94 × 10^−1^	1.07 × 10^−7^	1.01 × 10^−7^	3.84 × 10^−11^

Each value is expressed as mean ± SD. Means with different letters (a–e) are significantly different. Significant differences were determined by one-way ANOVA using the Fisher (F) test, and results were presented as *p*-values. The following are the abbreviations of the examined parameters: ABTS—radical scavenger capacity against 2,2′-azinobis(3-ethylbenzothiozoline)-6-sulfonic acid, ABTS^•+^; DPPH—radical scavenger capacity against 2,2-diphenyl-1-picrylhydrazyl radical, DPPH^•^; FRAP—ferric-reducing antioxidant power; NO—radical scavenger capacity against NO radical; ND—not detected.

## Data Availability

Data are contained within the article and Appendix A.

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
