# Peer review of "Polyamines in Edible and Medicinal Fungi from Serbia: A Novel Perspective on Neuroprotective Properties"

_jof, 2023, doi:10.3390/jof10010021_

Round 1

Reviewer 1 Report

Comments and Suggestions for Authors

Review for

 Title Novel Insights into the Neuroprotective Attributes of Polyamines in Selected Edible and Medicinal Fungi: A Pioneering Report 4

By Milena Rašeta 1,*, Marko Kebert 2, Jovana Mišković 3, Milana Rakić 3, Saša Kostić 2, Eleonora Čapelja 3, Maja Karaman 3

 This study aimed to investigate eight edible and medicinal fungi from Serbia, marking the initial investigation into the neuroprotective capabilities of Postia caesia, Clitocybe odora, Clitopilus prunulus, and Morchella elata. Neuroprotective  activity was examined using Ellman assay, while antioxidant capacity was tested by conducting 24 DPPH, NO, ABTS and FRAP tests.

Current title sounds like a review’ title, not an experimental paper

eight edible and medicinal fungi from Serbia’ should be present in the title

-----------------------------

 Fungi, which are  increasingly recognized as sources of natural antioxidants and AChE enzyme inhibitors, may thus  serve as potent neuroprotective agents. Previous studies have associated the anti-AChE and antioxidant activities of fungi mostly with polysaccharides and phenolic compounds, while other secondary metabolites such as polyamines (PAs) have been neglected.

while other secondary metabolites such as polyamines (PAs) have been neglected.

FULLY TRUE

----------------------------

 This study aimed to investigate eight  edible and medicinal fungi from Serbia, marking the initial investigation into the neuroprotective  capabilities of Postia caesia, Clitocybe odora, Clitopilus prunulus, and Morchella elata.

Why 8? Why these 8?

 Neuroprotective activity was examined using Ellman assay, while antioxidant capacity was tested by conducting  DPPH, NO, ABTS and FRAP tests.

Is there still an interest to run such antioxidant activity tests, so many compounds are antioxidant

Even the grass in my garden is antioxidant….

-----------------------

50,606 documents found

In a Scopus search with antioxidant AND activity

Peptides

Antioxidant and anti-HepG2 cell activities of a novel bioactive peptide from cowhide collagen in vitro

Xie, Z.Zhai, Y.Zhang, Y., ...Xiao, H.Song, Y.

Journal of Future Foods2024, 4(3), pp. 248–257

Polysaccharides

Effects of ultra-high pressure assisted extraction on the structure, antioxidant and hypolipidemic activities of Porphyra haitanensis polysaccharides

Zheng, M.Tian, X.Li, Z., ...Ni, H.Jiang, Z.

Food Chemistry2024, 437, 137856

Etc etc etc

-----------------------------

 However, it's important to recognize  that more research is required to identify the precise substances causing the reported inhibitory  effects.

Fully true, your study is very preliminary

Further studies should go deeper with one species at a time

--------------------

For example

·         1

Article  •  Open access

Postia caesia complex (Polyporales, Basidiomycota) in temperate Northern Hemisphere

Miettinen, O.Vlasák, J.Rivoire, B.Spirin, V.

Fungal Systematics and Evolution2018, 1, pp. 101–129

31

Citations

Show abstract

View at Publisher. Opens in a new tab.

Related documents

·         2

Article  •  Open access

Nomenclatural novelties in the Postia caesia complex

Papp, V.

Mycotaxon2014, 129(2), pp. 407–413

7

Citations

Show abstract

View at Publisher. Opens in a new tab.

Related documents

·         3

Article  •  Open access

Postia alni Niemelä & Vampola (Basidiomycota, Polyporales) - member of the problematic Postia caesia complex - has been found for the first time in Hungary

Papp, V.

Biodiversity Data Journal2014, 2(1), e1034

1

Citations

Show abstract

View at Publisher. Opens in a new tab.

Related documents

·         4

Article  •  Open access

Molecular variation in the Postia caesia complex

Yao, Y.-J.Pegler, D.N.Chase, M.W.

FEMS Microbiology Letters2005, 242(1), pp. 109–116

Author Response

Reviewer #1

  1. Reviewer #1 highlighted that our research is focused on exploring the neuroprotective potential of eight edible and medicinal fungi from Serbia, specifically investigating Postia caesia, Clitocybe odora, Clitopilus prunulus, and Morchella elata. The reviewer noted the examination of neuroprotective activity through the Ellman assay and the evaluation of antioxidant capacity using DPPH, NO, ABTS, and FRAP assays. Additionally, Reviewer #1 recommended adjusting the title to better reflect the experimental nature of the paper and incorporating all eight fungal names for precision.

Authors' response: Dear Reviewer #1, we would like to express our gratitude to you for your valuable feedback. We appreciate your thoughtful insights into our study, emphasizing its aim to explore the neuroprotective properties of eight edible and medicinal fungi from Serbia. We think about that to include names of all analyzed fungal species, but in that case title will be too large and not readable.
Therefore, it is not practical to incorporate all eight fungal names in the title as per your suggestion. Thank you for your time and consideration.

Abstract

  1. Reviewer #1 referenced our statement from the Abstract: "Fungi, which are increasingly recognized as sources of natural antioxidants and AChE enzyme inhibitors, may thus serve as potent neuroprotective agents. Previous studies have associated the anti-AChE and antioxidant activities of fungi mostly with polysaccharides and phenolic compounds, while other secondary metabolites such as polyamines (PAs) have been neglected," and concurred with our perspective.

Authors' response: Dear Reviewer #1, we appreciate your acknowledgment of our statement from the Abstract regarding the potential neuroprotective role of fungi. Your concurrence with our perspective reinforces the significance of our study, as it aims to shed light on the often-neglected secondary metabolites, particularly polyamines (PAs), in the context of anti-AChE and antioxidant activities. We believe that exploring these less-studied biomolecules will contribute valuable insights to the field.

Thank you for your continued engagement with our manuscript.

  1. Reviewer #1 inquired about the rationale behind our choice of these specific eight fungal species.

Authors' response: Dear Reviewer #1, we appreciate your inquiry regarding the selection of the eight fungal species in our study. The rationale behind our choice was multifaceted. Firstly, we aimed to explore previously uninvestigated species that are both edible and non-poisonous, emphasizing their relevance for human applications. In order to encompass diverse species with potential differences in bioactivities, we wanted to include different ecological (four terricolous – saprothrophic and mycorrhizal and four lignicolous) and systematic groups of fungi (Agaricales, Polyporales and Pezizales). Additionally, the availability of these species in sufficient quantity for analysis within the localities where we continuously explore the biodiversity and ecology of macrofungi, as well as their bioactivity, played important role in our research planning (species selection). Moreover, our research aims to provide novel insights to the scientific community by exploring previously biochemically understudied fungal species (Postia caesia, Clitocybe odora, Clitopilus prunulus, Lepista nuda, Morchella elata) as well as some of the well-known ones, but still poorly investigated from the point of polyamines content and neuroprotective activities (Ganoderma species and Cyclocybe aegerita).

Clarification on the selection of species was also added in the Manuscript, within the last paragraph of the Introduction (p. 3, l. 123-130, revised MS).

To underscore the pioneering nature of our study, we intentionally crafted the title as "Novel Insights into the Neuroprotective Attributes of Polyamines in Selected Edible and Medicinal Fungi: A Pioneering Report"

We hope this explanation clarifies our motivations and justifies the selection of the fungal species for our investigation.

  1. Reviewer #1 mentioned our statement from the Abstract: "Neuroprotective activity was examined using Ellman assay, while antioxidant capacity was tested by conducting DPPH, NO, ABTS and FRAP tests," and inquired whether there is continued interest in conducting antioxidant activity tests given the abundance of antioxidant compounds.

Authors' response: Dear Reviewer #1, we value your inquiry regarding the antioxidant activity tests and the referenced upcoming articles in your reviewer report, which discuss peptides and polysaccharides. As our title indicates, this paper aims to introduce new potential antioxidant and neuroprotective compounds, namely polyamines. While we acknowledge the general understanding of oxidative stress as the foundation of many diseases, it is crucial to first assess the potential of our samples to counteract oxidative stress. Moreover, it is important to underscore that polyamines from mushrooms are understudied and unexplored biomolecules, making them the focus of this study.

  1. Reviewer #1 referred to the final sentence in the Abstract, acknowledging that our findings are in the early stages, and suggested that future investigations should delve more profoundly into each species individually. Additionally, the reviewer provided a list of papers related to caesia, suggesting a focused and in-depth analysis of one species for future studies.

Authors' response: Dear Reviewer #1, we sincerely appreciate your valuable feedback. Our primary goal was to investigate various edible and medicinal fungal species, particularly those that have not been extensively studied before, focusing on edible and non-poisonous varieties with relevance for human applications. Following this study, we recognize the merit in narrowing our focus to Lepista nuda (inhibition of 99.94 ± 3.10%), Ganoderma applanatum (inhibition of 98.05 ± 0.83%), and Postia caesia (inhibition of 88.21 ± 4.76%) when discussing neuroprotective potential. Additionally, Postia caesia, Clitocybe odora, and Cyclocybe aegerita were identified as noteworthy suppliers of putrescine and spermidine polyamines.

We acknowledge the importance of more in-depth investigations into each fungal species individually. In our future research endeavors, we plan to conduct detailed studies on each species to provide a more comprehensive understanding of their bioactive compounds and potential applications. Your guidance is valuable, and we will incorporate a note in the revised manuscript to highlight our commitment to further exploration and detailed analysis of individual species. Thank you for your constructive feedback, and we acknowledge your suggestion and are grateful for it.

Reviewer 2 Report

Comments and Suggestions for Authors

Review

The manuscript entitled „ Novel Insights into the Neuroprotective Attributes of Polyamines in Selected Edible and Medicinal Fungi: A Pioneering Report” presents the results of some chemical and biological activity experiments made on certain edible and/or medicinal mushrooms native to Serbia. The selected topic is notable, which can capture the attention of scientific audience, however the term pioneering seems to be exaggerated.

The manuscript itself is generally well written, structured in a logical way, however there are some questions to be addressed prior a potential publication:

·        What was the selection criteria for the examined mushroom species?

·        Did you performed preliminary research to determine which solvent(s) were the best for extraction of mushroom species? Depending on the polarity of the solvent used, the chemical profile of the extract can greatly vary containing more or less compounds responsible for the observed pharmacological activity.

·        Line 134: “The fruiting bodies of the selected fungal species were sliced into small fragments and subsequently subjected to hot air drying in an oven (Universal oven UF55 – Memmert GmbH + Co. KG, Germany) at 45 °C for 24 hours.” Why did not you apply freeze drying? Lyophilization is one the most suitable drying methods to preserve the characteristic bioactive compounds of mushrooms avoiding the potential enzymatic processes leading to the decomposition of certain fungal metabolites.

·        The flavonoid content of mushrooms is a problematic issue since their regular presence in mushrooms has not been unequivocally proved. I am not fully convinced about the flavonoid content of fungal species.

·        It would be good to see the contribution of certain isolated compounds to the observed biological activity.

·        Which of the antioxidant assay you consider the most relevant in this study?

·        Carbon and nitrogen elemental analysis is basically nice but with lower relevance in this study.

·        What was the gradient used for HPLC? Please specify.

·        Line 102: “The health benefits of polyphenols and carbohydrates from fungi are primarily linked to their antioxidant properties.” This is not necessarily true since they can have other important biological activities.

·        The species names in figures 1 and 3 are practically unreadable. Please increase the font size.

·        Line 166: it is said that methanol was used to make fungal extracts, then in line 168 it was mentioned that 80% methanol solution was used. Which one is correct?

·        Instead of Lentinus edodes use Lentinellus edodes, the current scientific name of this species.

·        Line 633: the phrase “diverse range of fungi” seems to be inappropriate.

Comments on the Quality of English Language

English basically fine, check for typographical errors

Author Response

Reviewer #2

  1. Reviewer #2 noted that our research article, titled "Novel Insights into the Neuroprotective Attributes of Polyamines in Selected Edible and Medicinal Fungi: A Pioneering Report," highlights the outcomes of chemical and biological activity experiments conducted on specific edible and/or medicinal mushrooms indigenous to Serbia. While acknowledging the significance of the chosen topic in attracting scientific interest, the term "pioneering" appears to be overstated. Overall, the manuscript is well-written and logically structured, but there are some queries that should be addressed before considering publication.

Authors' response: Dear Reviewer #2, we appreciate your thoughtful comments and suggestions. Regarding the term "pioneering" in our title, it is grounded in the lack of existing data on polyamine content for all eight fungal species analyzed in our study. Additionally, the anti-acetylcholinesterase and antioxidant activities of most of these species remain largely unexplored. We believe that our research contributes valuable insights to this underexplored area, justifying the use of the term "pioneering." Your feedback is crucial, and we will ensure to address any concerns you have raised in our revisions. Thank you for your valuable input.

  1. Reviewer #2 inquired about the criteria employed for the selection of the eight species.

Authors' response: Dear Reviewer #2, we appreciate your inquiry regarding the selection of the eight fungal species in our study. The rationale behind our choice was multifaceted. Firstly, we aimed to explore previously uninvestigated species that are both edible and non-poisonous, emphasizing their relevance for human applications. In order to encompass diverse species with potential differences in bioactivities, we wanted to include different ecological (four terricolous – saprothrophic and mycorrhizal and four lignicolous) and systematic groups of fungi (Agaricales, Polyporales and Pezizales). Additionally, the availability of these species in sufficient quantity for analysis within the localities where we continuously explore the biodiversity and ecology of macrofungi, as well as their bioactivity, played important role in our research planning (species selection). Moreover, our research aims to provide novel insights to the scientific community by exploring previously biochemically understudied fungal species (Postia caesia, Clitocybe odora, Clitopilus prunulus, Lepista nuda, Morchella elata) as well as some of the well-known ones, but still poorly investigated from the point of polyamines content and neuroprotective activities (Ganoderma species and Cyclocybe aegerita).

Clarification on the selection of species was also added in the Manuscript, within the last paragraph of the Introduction (p. 3, l. 123-130, revised MS).

  1. Reviewer #2 inquired about whether preliminary research was conducted to determine the most effective solvent(s) for extracting mushroom species. The choice of solvent polarity can significantly impact the chemical profile of the extract, influencing the presence of compounds responsible for observed pharmacological activity.

Authors' response: Dear Reviewer #2, we appreciate your inquiry regarding the solvent selection for extraction. We considered the findings of the article by Gąsecka et al. (2016) where 80% MeOH was identified as an effective solvent for extracting phenolic compounds (4-hydroxybenzoic, 2,5-dihydroxybenzoic, ferulic, p-coumaric, protocatechuic, t-cinnamic, and vanillic acids) and flavonoids (naringenin). This solvent choice aligns with our study's focus on measuring antioxidant potential, quantifying total phenolic and flavonoid compounds, and testing the neuroprotective potential using the Ellman assay. Additionally, we have found this type of extract to be a valuable source of phenolics, essential secondary metabolites in mushrooms, as highlighted in some of our upcoming papers.

Used reference:

Gąsecka M., Mleczek, M., Siwulski, M., & Niedzielski, P. (2016). Phenolic composition and antioxidant properties of Pleurotus ostreatus and Pleurotus eryngii enriched with selenium and zinc. European Food Research and Technology, 242, 723-732.

  1. Reviewer #2 referred to our statement in Line 134, where we mentioned that the fruiting bodies of the chosen fungal species were sliced into small fragments and subjected to hot air drying in an oven at 45 °C for 24 hours. The reviewer raised a question about why freeze drying was not employed, considering that lyophilization is considered one of the most suitable drying methods for preserving the characteristic bioactive compounds of mushrooms and preventing potential enzymatic processes that could lead to the decomposition of certain fungal metabolites.

Authors' response: Dear Reviewer #2, thank you on your comment regarding method of drying our samples. Yes, we know that and also we often use lyophilization as method for  drying but we are aware that lyophilization is the best way to preserve bioactive substances in mushrooms, but dried mushrooms are widely used since it's a more convenient and less expensive way of preserving mushrooms.

  1. Reviewer #2 has expressed concerns about the flavonoid content of mushrooms, highlighting the challenge of unequivocally proving their regular presence in these fungi. Additionally, the reviewer remains unconvinced about the reported flavonoid content in fungal species.

Authors' response: Dear Reviewer #2, we appreciate your comment regarding the quantification of flavonoid compounds in mushrooms. Our research group has previously published numerous papers where we quantified various compounds, including phenolics such as phenolic acids and flavonoids (numerical references from the text: 3, 8-10, and 50). For example: Rašeta et al. 2020a (Ref 8 in this paper) reported the highest phenolic acid content in extracts of four Ganoderma species (G. applanaum, G. lucisum, G. pfeifferi and G. resinaceum) using LC-MS/MS. Flavonoid and coumarin compounds were exclusively detected in EtOH extracts of G. pfeifferi. In our study, we identified and quantified two flavonoid compounds (chrysoeriol and isorhamnetin) in G. pfeifferi extracts for the first time in Ganoderma species. We employed an unspecific colorimetric method (Chang et al., 2002) for a rapid total flavonoid content screening in our extracts as an initial examination. While literature suggests skepticism due to the lack of flavonoid biosynthetic pathway sequences in mushrooms, recent research (Mohanta, 2020) asserts that mushrooms contain the genomic architecture needed for flavonoid biosynthesis, supporting the determination and quantification of flavonoids in mushrooms. According to the literature findings (Gil-Ramírez et al., 2016), the results we obtained may raise suspicions because mushrooms lack sequences encoding key enzymes in the flavonoid biosynthetic pathway. Additionally, there is no significant absorption observed in fruiting bodies cultivated in flavonoid-enriched substrates or in mycelia grown in flavonoid-supplemented lab media. Consequently, the use of spectrophotometric methods for determining 'total flavonoids' in fungal samples is discouraged (Gil-Ramírez et al., 2016). Conversely, a recent study by Mohanta (2020) reported that mushrooms, being the fruiting body of fungi, possess the entire genomic architecture of their genome. This indicates that mushrooms also contain enzymes associated with the biosynthesis of flavonoids, allowing for their determination and quantification in mushrooms.

Used references:

Chang, C. C., Yang, M. H., Wen, H. M., Chern, J. C. (2002) Estimation of total flavonoid content in propolis by two complementary colorimetric methods. Journal of Food and Drug Analysis. 10: 178-182.

Gil-Ramírez, A., Pavo-Caballero, C., Baeza, E., Baenas, N., Garcia-Viguera, C., Marín, F. R., Soler-Rivas, S. (2016) Mushrooms do not contain flavonoids. Journal of Functional Foods. 25: 1-13.

Mohanta, T. K. (2020) Fungi contain genes associated with flavonoid biosynthesis pathway. Journal of Functional Foods. 68: 103910.

We trust that our clarification addresses Your concerns.

Top of Form

Top of Form

  1. Reviewer #2 suggested that it would be good to see the contribution of certain isolated compounds to the observed biological activity.

Authors' response: Dear Reviewer #2, Thank you for your valuable suggestion. We appreciate your interest in understanding the specific contribution of isolated compounds to the observed biological activity. In our research article we have focused on providing statistical significance (we applied: Pearson’s correlation coefficient and PCA analysis) for quantified compounds, particularly polyamines, as well as total phenolic, total flavonoid, and total protein levels in relation to the analyzed antioxidant and neuroprotective activities.

We are pleased to share some noteworthy results from our study, emphasizing the neuroprotective potential of Postia caesia. Specifically, P. caesia exhibited significant acetylcholinesterase (AChE) inhibition (88.21 ± 4.76%), which was primarily associated with elevated spermidine (SPD) levels (62.98 ± 3.19 mg/kg d.w.) and putrescine (PUT) levels (55.87 ± 3.16 mg/kg d.w.). Moreover, we observed a general correlation between AChE inhibition and the levels of SPD and PUT. Additionally, our results highlight that the scavenging capacity assays demonstrated the highest activity in association with total phenolic compounds. Furthermore, a noteworthy correlation was identified between total flavonoids and total proteins concerning antioxidant activity.

We hope this additional information addresses your inquiry adequately. If you have any further questions or suggestions, please feel free to let us know.

  1. Reviewer #2 inquired about our perspective on which antioxidant assay we deem the most relevant in this study.

Authors' response: Dear Reviewer #2, we appreciate your inquiry regarding the choice of the most relevant antioxidant assay in our study. Table 2 highlights that G. applanatum and L. nuda exhibited the most robust antioxidant activities across various assays. Specifically, G. applanatum demonstrated superior scavenging of ABTS radicals and reduction in the FRAP assay, while L. nuda displayed the highest activity in the DPPH and NO assays, and notable FRAP activity, ranking as the second most potent species after G. applanatum. This sequence of antioxidant activities was consistent across different pH values during the assays, considering the protonation of compounds at low pH and their role as significant hydrogen donors in neutralizing free radicals.

Our findings align with studies, such as Huang et al. (2005), which emphasized that ET-BASED ASSAYS (electron-transfer Based assays), including TEAC, FRAP, and DPPH, involve similar redox reactions based on the electron transfer (ET) mechanism. The positive correlation observed between ABTS, total phenolic compounds (TPC), and FRAP supports the similarity in the underlying redox reactions. Notably, all antioxidant assays, except NO assay, focus on measuring total antioxidant capacity, while the NO assay is more specific and closely linked to neuroprotective properties. Our extracts exhibited potential in scavenging NO radicals, as indicated in Table 2.

Furthermore, Tripathi et al. (2020) highlighted in their review the role of NO in various brain disorders and emphasized the contribution of NO to nitrosative stress in the nervous system. Our study, therefore, considers the relevance of NO assay in providing insights into the neuroprotective properties of the analyzed fungal extracts.

In addressing your query in our conclusion, we'd like to emphasize that the selection of these assays was primarily driven by the dual purpose of investigating the total antioxidant capacity (ABTS, DPPH, and FRAP assays) and assessing the extracts' ability to scavenge NO radicals. NO is recognized as one of the crucial signalling molecules in the brain, playing a pivotal role in the genesis of various brain-related disorders. Its impact can be dual, acting both protectively/constitutively and destructively/toxically, contingent upon its regulation/production and interactions with different molecules within the cell. This dual nature has led to the characterization of NO as a "double-edged sword."

Top of Form

Used references:

Tripathi, M.K., Kartawy, M., & Amal, H. (2020). The role of nitric oxide in brain disorders: Autism spectrum disorder and other psychiatric, neurological, and neurodegenerative disorders. Redox Biology, 34, 101567.

Huang, D., Ou, B., & Prior, R.L. (2005). The chemistry behind antioxidant capacity assays. Journal of Agricultural and Food Chemistry, 53 6, 1841-1856.

  1. Reviewer #2 commented that carbon and nitrogen elemental analysis is basically nice but with lower relevance in this study.

Authors' response: Dear Reviewer #2, we appreciate your comment on the relevance of the Carbon and nitrogen elemental analysis. We concur with your assessment that this analysis may have limited significance in this study. However, it was deemed important as it allowed us to measure nitrogen content, facilitating the recalculation of total protein content—one of the essential parameters in assessing the mycochemical composition of the analyzed species.

  1. Reviewer #2 asked that we explain and specify what was the gradient used for HPLC analysis.

Authors' response: Dear Reviewer #2, we appreciate your inquiry regarding the HPLC gradient used in our study. We utilized a specific gradient, and for clarity, we will provide detailed specifications in the revised manuscript (p. 4, l. 156-162, revised MS). Also, there is copy of the text:

„The following five steps of programmed acetonitrile:water (v/v) gradient were applied: 60 to 70% of acetonitrile in 5.5 min, 70 to 80% in 1.5 min, 80 to 100% in 2 min, 100 to 100% in 2 min, 100 to 70% in 2 min and 70 to 60% in 2 min at a flow rate of 1.5 mL/min. Eluted compounds were identified using a prominence fluorescence detector RF-20A (365 nm excitation, 510 nm emission). A postrun programme Lab Solutions was used to integrate the areas of the peaks originated from dansylated polyamines.“

  1. Reviewer #2 referred to our statement from line 102: "The health benefits of polyphenols and carbohydrates from fungi are primarily linked to their antioxidant properties." and pointed out that this might not be entirely accurate, as these compounds could have other significant biological activities.

Authors' response: Dear Reviewer #2, we value your input on this issue, and we have revised the specified sentence and the one following it accordingly (p. 3, l. 106-109, revised MS).

“The health benefits of polyphenols and carbohydrates derived from fungi are predominantly associated with their antioxidant properties, among others biological activities. Nevertheless, emerging studies indicate that these compounds might also contribute to promoting neuroprotection [8, 20–22].”

  1. Reviewer #2 pointed out that the species names in Figures 1 and 3 are nearly illegible and suggested enlarging the font size.

Authors' response: Dear Reviewer #2, we appreciate your feedback on the legibility of fungal species names in Figures 1 and 3. We have examined the font size and believe it is adequate. We remain hopeful that the readability will be improved in the printed version of our paper.

  1. Reviewer #2 commented that in the line 166: we said that methanol was used to make fungal extracts, and then in line 168 we mentioned that 80% methanol solution was used. Which one is correct?

Authors' response: Dear Reviewer #2, thank you for bringing this to our attention. We acknowledge the oversight regarding the use of 80% MeOH as a solvent for extraction. In the revised version of the manuscript, we have corrected this error (p. 4, l. 179, revised MS).

  1. Reviewer #2 suggests that instead of Lentinus edodes we should use Lentinellus edodes, the current scientific name of this species.

Authors' response: Dear Reviewer #2, we appreciate your suggestion. However, we utilized the current names provided by Index Fungorum (https://www.indexfungorum.org/names/NamesRecord.asp?RecordID=287561) and MycoBank (https://www.mycobank.org/page/Name%20details%20page/field/Mycobank%20%23/316467). According to both sources, Lentinula edodes is the current name for this species. We will make the necessary corrections throughout the entire manuscript (p. 3, l. 116, revised MS and p. 7, l. 288, revised MS).

  1. Reviewer #2 commented that in the line 633 we used the phrase “diverse range of fungi” and he think that it is inappropriate.
  • Top of Form

Authors' response: Dear Reviewer #2, thank you for your comment on the phrase "diverse range of fungi" used in line 633. We acknowledge your feedback, and in response, we will reconsider and modify the wording for clarity and appropriateness (p. 15, l. 654, revised MS).

Round 2

Reviewer 2 Report

Comments and Suggestions for Authors

Dear Authors,

Thank you for the responses to my questions. Most of the suggestions have been considered, and thus the manuscript has been improved. However, in some points I do maintain my opinion. I still consider that it is not a pioneering report. I do not want to polemize here, and transform the review into a linguistic topic, but according to Cambridge Dictionary, definition of pioneering is: “using ideas and methods that have never been used before”. And I think this is not the case of this manuscript.

The other issue is the flavonoid content. Here I would prefer an HPLC analysis rather than the colorimetric assay used.

Regarding the comment no. 10 („The health benefits of polyphenols and carbohydrates from fungi are primarily linked to their antioxidant properties.") the primary and most important pharmacological feature of fungal polysaccharides is not the antioxidative property, but the immunomodulatory activity.

Comment no. 11: As far as I see, the names of fungal species in the figures are not readable.

Comments on the Quality of English Language

The language of the manuscript is generally fine.

Author Response

Reviewer #2

  1. Reviewer #2 expressed appreciation for our responses to their queries, acknowledging that most of the suggestions have been taken into account and that the revised manuscript has shown improvement. Nevertheless, Reviewer #2 has requested further clarification on certain aspects.

Authors' response: Dear Reviewer #2, we value your insightful comments and suggestions, and we are grateful as they have contributed to the enhancement of our paper.

  1. Reviewer #2 observed that the term "pioneering" in our research article, titled "Novel Insights into the Neuroprotective Attributes of Polyamines in Selected Edible and Medicinal Fungi: A Pioneering Report" seems exaggerated. Despite our clarification, Reviewer #2 maintains the opinion that the report does not qualify as pioneering.

Authors' response: Dear Reviewer #2, we appreciate your thoughtful comments and suggestions, and regarding them we changed title of our paper as: “Polyamines in Edible and Medicinal Fungi from Serbia: A Novel Perspective on Neuroprotective Properties” (p. 1, l. 2-3, revised MS)

  1. Reviewer #2 raised a concern about the flavonoid content and recommended that we consider utilizing HPLC analysis for our samples/extracts instead of relying on the colorimetric assay.

Authors' response: Dear Reviewer #2, we acknowledge your inquiry regarding the most precise analysis for quantifying flavonoid compounds. This study's primary focus was to explore the often-overlooked secondary metabolites, polyamines, in the context of anti-AChE and antioxidant activities. We believe that delving into these less-explored biomolecules will provide valuable insights into the secondary metabolites of fungal species.

While phenolic acids are well-known antioxidants, and flavonoids are also important in this regard, we recognize that precise analysis of both compounds requires HPLC techniques. In this study, our objective was to quantify total phenolic and flavonoid content and examine the correlation between them and the analyzed antioxidant activity. However, the main emphasis was on quantifying polyamines to explore the potential of these species as novel sources of neuroprotective agents.

In our previous publications, we employed LC-MS/MS analysis, a method established by our colleagues from the Department of Chemistry, Biochemistry, and Environmental Protection (Faculty of Sciences, University of Novi Sad, Serbia), as evidenced by Orčić et al. (2014). Our future research endeavors include quantifying phenolic and flavonoid compounds using the LC-MS/MS technique to investigate antidiabetic and antiproliferative potential, among other activities, and to assess whether phenolic and flavonoid compounds impact these activities.

These articles represent some of our previously published works wherein we quantified phenolics and flavonoids using the LC-MS/MS technique:

  1. Krsmanović, N.; Rašeta, M.; Mišković, J.; Bekvalac, K.; Bogavac, M.; Karaman, M.; Isikhuemhen, O.S. Effects of UV stress in promoting antioxidant activities in fungal species Тrametes versicolor (L.) Lloyd and Flammulina velutipes (Curtis) Singer. Antioxidants 2023, 12(2), 302.
  2. Mišković, J.; Karaman, M.; Rašeta, M.; Krsmanović, N.; Berežni, S.; Jakovljević, D.; Piattoni, F.; Zambonelli, A.; Gargano, M.L.; Venturella, G. Comparison of two Schizophyllum commune strains in production of acetylcholinesterase inhibitors and antioxidants from submerged cultivation. J. Fungi 2021, 7(2), 115.
  3. Rašeta, M.; Karaman, M.; Jakšić, M.; Šibul, F.; Kebert, M.; Novaković, A.; Popović, M. Mineral composition, antioxidant and cytotoxic biopotentials of wild‐growing Ganoderma species (Serbia): G. lucidum (Curtis) P. Karst vs. G. applanatum (Pers.) Pat. Int. J. Food Sci. Technol, 2016, 51, 2583–2590.
  4. Rašeta, M.; Popović, M.; Čapo, I.; Stilinović, N.; Vukmirović, S.; Karaman, M. Antidiabetic effect of two different Ganoderma species tested in alloxan diabetic rats. RSC Adv. 2020, 10, 10382–10393.
  5. Rašeta, M.; Popović, M.; Beara, I.; Šibul, F.; Zengin, G.; Krstić, S.; Karaman, M. Anti-inflammatory, antioxidant and enzyme inhibition activities in correlation with mycochemical profile of selected indigenous Ganoderma spp. from Balkan region (Serbia). Chem. Biodivers 2020, 17, e2000828.

  1. Reviewer #2 pointed out in relation to comment no. 10 from the previous revision ("The health benefits of polyphenols and carbohydrates from fungi are primarily linked to their antioxidant properties") that the fundamental and crucial pharmacological characteristic of fungal polysaccharides is not their antioxidative nature, but rather their immunomodulatory activity.

Authors' response: Dear Reviewer #2, we appreciate your feedback concerning the primary and most crucial pharmacological feature of fungal polysaccharides being associated with their immunomodulatory properties. We will revise the sentence accordingly:

“The advantageous effects of fungal polyphenols are mainly connected to their antioxidant attributes, whereas carbohydrates obtained from fungi exhibit immunomodulatory activity, along with various other biological activities.” (p. 3, l. 107-109, revised MS).

  1. Reviewer #2 highlighted, in reference to comment 11 from the previous revision, that the names of fungal species in Figures 1 and 3 are not legible.

Authors' response: Dear Reviewer #2, we acknowledge your feedback on the legibility of fungal names in Figures 1 and 3. In response, we have generated new figures with larger font sizes for improved readability of fungal species names.

Figure 1.

Figure 3. (Please refer to the revised figures in the attached Word file.)
